# Despite Low Obesity Rates, Body Mass Index Under-Estimated Obesity among Russian Police Officers When Compared to Body Fat Percentage

**DOI:** 10.3390/ijerph17061937

**Published:** 2020-03-16

**Authors:** Katie M. Heinrich, Konstantin G. Gurevich, Anna N. Arkhangelskaia, Oleg P. Karazhelyaskov, Walker S. C. Poston

**Affiliations:** 1Department of Kinesiology, Kansas State University, Manhattan, KS 66506, USA; 2Moscow State University of Medicine and Dentistry (MSUMD), Moscow 127473, Russia; kgurevich@mail.ru (K.G.G.); cattiva@list.ru (A.N.A.); olegkara@mail.ru (O.P.K.); 3Research Institute of Health Organization and Medical Management of the Department of Public Health, Moscow 115184, Russia; 4NDRI-USA, New York, NY 10001, USA; poston@ndri-usa.org

**Keywords:** bioelectrical impedance, body composition, first responders, law enforcement officers, waist circumference

## Abstract

In some countries, obesity rates among police officers are higher than the general public, despite physically demanding jobs. Obesity rates based on body mass index (BMI) may lack accuracy as BMI does not directly address body composition. Since data are lacking for obesity rates among Russian police officers, this study documented and compared officer obesity rates to the adult Russian population and compared the accuracy of body mass index (BMI) for obesity classification to two direct measures of body composition. Moscow region police officers (*N* = 182, 84% men) underwent height, weight, waist circumference (WC), and body fat percentage (BF%) bioelectrical impedance measurements during annual medical examinations. BMI-defined obesity rates were 4.6% for men and 17.2% for women, which were >3 and >1.8 times lower than Russian adults, respectively. WC-defined obesity rates were similar to BMI (3.3% for men and 10.3% for women), but BF%-defined obesity rates were much higher (22.2% for men and 55.2% for women). Although obesity rates were lower than those found among police officers in other countries, BMI alone was not a particularly accurate method for classifying weight status among Russian police officers.

## 1. Introduction

Policing is an important worldwide government function for citizen protection. The job-demands of police officers require them to have higher levels of fitness than most other occupations [1] as they must be prepared to respond to physical challenges on a regular basis (e.g., chase suspects, subdue resisting arrestees) [2]. In addition, weight status is a factor in the hiring process to ensure adequate job performance capacity [3]. However, objective monitoring of United States (US) police officers has found the majority of their shifts are spent in sedentary activities [4,5], likely contributing to police officers having greater obesity rates than the general public [6]. For example rates of overweight and obesity combined based on body mass index (BMI) are up to 80% among Iowa, US police officers [3].

Yet, BMI has been known to overestimate obesity among muscular individuals and underestimate obesity among those with low muscle mass due to its reliance on height and weight rather directly assessing body composition [7]. Among US police officers, BMI has incorrectly identified participants as obese 48.8% of the time [3]. Surprisingly, the bulk of misclassifications were not due to increased muscle mass, but actually due to misclassifying officers as normal or overweight instead of obese when compared to body fat percentage (BF%).

It is also of concern that obesity is a cardiovascular disease risk factor, for which police officers already have an increased risk [8]. As there are multiple ways to measure obesity, waist circumference (WC) is a recommended measure of central adiposity (i.e., excess fat tissue) to help indicate increased cardiovascular disease risk [9]. WC has been compared to both BMI and BF% among US police officers [10], female firefighters [11], and military personnel [12], as well as Russian firefighters [13]. All three measures (i.e., BMI, BF%, and WC) have been highly correlated in US police officers [10].

As the largest country of the former Soviet Union, Russia is diverse, rich with resources, and has a rapidly changing food economy [14] where diets include excess amounts of saturated fats, sugar, and salt, along with insufficient consumption of fruits and vegetables [15]. As a result, and similar to much of the rest of the world, the Russian Federation (Russia) population has experienced increases in obesity rates [14,16]. Although Russian women have higher obesity rates, rates among men are increasing at a faster pace [16]. This is concerning because those with obesity are more likely to have co-morbid health conditions such as hypertension, diabetes, and high triglycerides [16], and the majority of health coverage funding is from the public sector (61%) [17].

Currently, there no studies that have examined the prevalence of obesity among police officers in Russia. In addition, no data exist determining the accuracy of BMI for classifying weight status in this group. The purpose of this observational study was to document rates of obesity and compare age-standardized sex-specific rates for officers to the adult population in Russia. In addition, we examined the accuracy of BMI-based obesity classification when compared to BF% and WC measurements.

## 2. Materials and Methods

This study received ethical approval from the Moscow State University of Medicine and Dentistry Ethics Committee (#02-13 from 21.02.2013) and the National Development and Research Institutes, Inc. approved a data use/sharing agreement. Participants included Moscow region police officers undergoing mandatory annual medical examinations in March and April 2015. Police officers were required to complete an annual medical evaluation that was scheduled in the morning between 0800–0900 h. Those who slept less than 7 h the previous night or had eaten prior to their appointment were excluded from the study. Each participant (*N* = 182) completed written informed consent prior to participation.

Participants were instructed to have an adequate night’s sleep and not consume any food or drinks prior to testing. Participants indicated their age (years) and sex (men or women). Measures included height (cm), weight (kg), and WC (cm) [9]. The Medas 1.0 (SRC Medas, Moscow, Russia), a whole body tetrapolar bioelectrical impedance device that used a 50 kHZ frequency, was used to assess body composition (fat mass in kg) with participants in a supine position using conventional tetrapolar electrode (i.e., Ag/AgCl Schiller bioadhesive electrodes) placement [18]. BMI (kg/m^2^) was calculated from measured height and weight and officers were categorized as underweight (<18.5 kg/m^2^), normal weight (18.5–24.9 kg/m^2^), overweight (25–29.9 kg/m^2^), or obese (≥30 kg/m^2^) [19]. BF% was calculated by dividing fat mass by total body mass and multiplying by 100.

Statistical analyses were completed with SPSS version 25 (IBM, Armonk, NY, USA). First, crude prevalence of overweight and obesity combined (BMI ≥ 25), obesity (BMI ≥ 30) and obesity based on BF% (≥25% for men and ≥30% for women) [20], and WC (>102 cm for men and >88 cm for women) [9] were computed and the bivariate correlations examined. Next, age-standardized estimates were computed for each sex [21] and compared to national obesity rates based on BMI [22]. Finally, rates of false positive and false negative classifications, as well as overall accuracy of BMI-based obesity classification was evaluated by comparison with BF%- and WC-determined obesity status. Statistical significance was set at *p* ≤ 0.05.

## 3. Results

### 3.1. Descriptive Results

Of the 243 eligible police officers solicited during annual medicals, 182 consented to participate (74.9%). Participants included 153 men (84.1%) and 29 women (15.9%), with an average age of 27.1 ± 7.3 years (range = 18–48 years). Mean BMI, WC, and BF% values were, respectively, 25.7 ± 8.4 kg/m^2^, 86.4 ± 7.1 cm, and 21.0 ± 6.0% for men and 25.6 ± 5.4 kg/m^2^, 82.2 ± 12.7 cm, and 33.1 ± 7.5% for women. BMI weight status categories showed that 53.6% (*n* = 82) of men were normal weight, while 41.8% (*n* = 64) were overweight, and 4.6% (*n* = 7) were obese. For women 58.6% (*n* = 17) were normal weight, 24.1% (*n* = 7) were overweight, and 17.2% (*n* = 5) were obese using BMI.

### 3.2. Comparisons Between Obesity Measurements

#### 3.2.1. Correlation Analyses and Obesity Prevalence Rates by Measurement

Bi-variate correlation analyses for BMI, WC, and BF% demonstrated medium-to-large associations for men (r_BMI-WC_ = 0.46, *p* < 0.01; r_BMI-BF%_ = 0.68, *p* < 0.01; r_WC-BF%_ = 0.65, *p* < 0.01) and large associations for women (r_BMI-WC_ = 0.93, *p* < 0.01; r_BMI-BF%_ = 0.92, *p* < 0.01; r_WC-BF%_ = 0.89, *p* < 0.01). The combined prevalence rates of BMI-based overweight and obesity were 46.4% (*n* = 71) for men and 41.4% (*n* = 12) for women. Prevalence of BMI-defined obesity (i.e., 4.6% for men and 17.2% for women) and WC-defined obesity prevalence (3.3%, *n* = 5 for men and 10.3%, *n* = 3 for women) were much lower than BF%-defined obesity prevalence (22.2%, *n* = 34 for men and 55.2%, *n* = 16 for women). Figure 1 and Figure 2 show the crude and age-standardized estimates for BMI-defined obesity among men and women Russian police officers and the most recent obesity prevalence rates for Russian adult men and women.

#### 3.2.2. Rates of False Positives and False Negatives for Men

Rates of false positive and false negative BMI-based obesity classification, when compared with WC- and BF%-determined obesity status are shown in Table 1 for men. False positive rates (i.e., misclassified as obese using BMI) for non-obese men were 3.4% for WC and 2.5% for BF%, while false negative rates (i.e., misclassified as non-obese using BMI) for obese men were 0.6% for WC and 11.8% for BF%. BMI was more accurate for correctly classifying participant obesity status when WC (94.8%) was used for the standard, as compared to BF% (78.4%).

#### 3.2.3. Rates of False Positives and False Negatives for Women

Table 2 displays the rates of false positive and false negative for comparing WC and BF% to BMI-defined obesity for women police officers. False positive rates (i.e., misclassified as obese using BMI) for non-obese women were 7.7% for WC and 0% for BF%, while false negative rates (i.e., misclassified as non-obese using BMI) for obese women were 0% for WC and 68.8% for BF%. Similar to men, BMI was more accurate for correctly classifying participant obesity status when WC (93.1%) was used for the standard as compared to BF% (62.1%).

## 4. Discussion

This is the first study to provide data on the accuracy of BMI, as well as obesity rates, among Russian police officers. We compared age-standardized sex-specific obesity rates to the adult Russian population and examined the accuracy of BMI-based obesity classification in comparison to BF% and WC. Specifically, we found age-standardized obesity prevalence rates, when compared to the adult Russian population, were >3 times and >1.75 times lower for men and women police officers, respectively. We also found that BMI-defined obesity was relatively accurate for men and women police officers (both > 93%) when compared to WC-defined obesity, resulting in more false positives than false negatives. This suggests that using BMI for obesity classification would overestimate obesity rates when compared with WC.

However, BMI-defined obesity was less accurate when compared to BF%-determined obesity, with overall diagnostic accuracy of 78.4% and 62.1% for men and women police officers, respectively. In addition, rates of false positives were negligible while the rates of false negatives were much higher, suggesting that BMI underestimates obesity rates when compared with BF% in this population. Thus, when measuring obesity among Russian police officers, it may be helpful to directly assess BF% rather than using estimation methods such as BMI and WC.

As the first study examining obesity prevalence among Russian police officers, we did not find that participants’ obesity rates were as high as Russian adults at large [14], or male Russian firefighters [13], nor were they similar to data from other countries showing higher obesity rates among police officers than the general public [6]. Despite lower obesity rates among our participants, we did not find that BMI overestimated obesity rates (i.e., high false positives) [7], particularly when compared to BF%. In fact, the low false positive rates were similar to previous research with male Russian firefighters [13]. However, our finding that BMI-determined obesity was likely to underestimate obesity (i.e., higher rates of false negatives) was similar to previous research among US police officers and Russian firefighters [3,13], as well as our previous research among female firefighters that found obesity based on measured BMI was significantly lower than that based on BF% (57.1%–66.7% lower) [11].

Although we did not directly compare BMI-based obesity rates between male and female police officers, our prevalence rates were numerically higher among females. This finding is in contrast to research from Quebec, Canada where 21.1% of male and only 7.3% of female police officers were obese according to BMI [23]. Additionally, research from New York, USA, found 50.7% of male and only 21.3% of female police officers were obese according to BMI [10]. It may be that additional lifestyle factors not measured in our study accounted for the different prevalence rates by sex [8]. For example, additional research in Quebec found that 12% of female officers (*N* = 41) were obese and 62% reported being physically inactive [24].

Study strengths include the use of multiple direct measurements of body composition for participants. Of note, our BF% method has been validated against other standardized measures [25] and all measures were administered by trained professionals. In addition, our study provides body composition data among a previously unstudied group, i.e., police officers in Russia. Limitations include that participants were recruited from a convenience sample of police officers undergoing annual medical evaluations. Thus, results from this study cannot be generalized to Russian police officers in general. It is possible that those reporting for the evaluations were not representative of all officers due to their much lower obesity rates overall when compared to the general Russian adult population. Since only BMI data were available for the Russian population, we were unable to compare obesity rates based on BF% and WC with our sample.

## 5. Conclusions

Based on our study findings, we recommend that BMI-based obesity is most accurate when compared to WC measurements. However, to more accurately capture the full extent of obesity among police officers, including BF%-based measurements will increase the likelihood of obesity underestimation. As the Russian Federation currently has no policies regarding obesity among police officers, our data provide a starting point for discussions about the subject. In addition, future research among Russian police officers should incorporate self-reported and objective assessments of physical activity levels to determine their relationship with obesity status.

## Figures and Tables

**Figure 1 ijerph-17-01937-f001:**
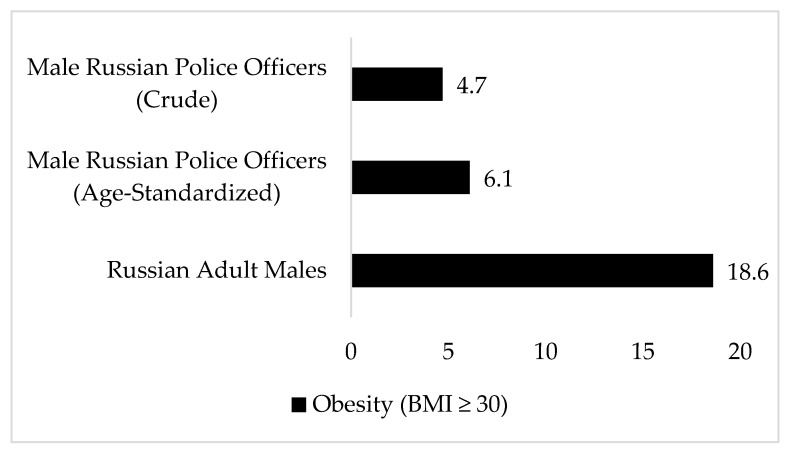
Age-standardized prevalence of obesity (BMI ≥30 kg/m^2^) in male Russian police officers compared with Russian adults. Russian police officer rates were age-standardized using 2014 age-stratified population estimates for Russian males provided in the Central Intelligence Agency World Factbook for Russia [21]; Russian male obesity estimates are from the World Health Organization [22].

**Figure 2 ijerph-17-01937-f002:**
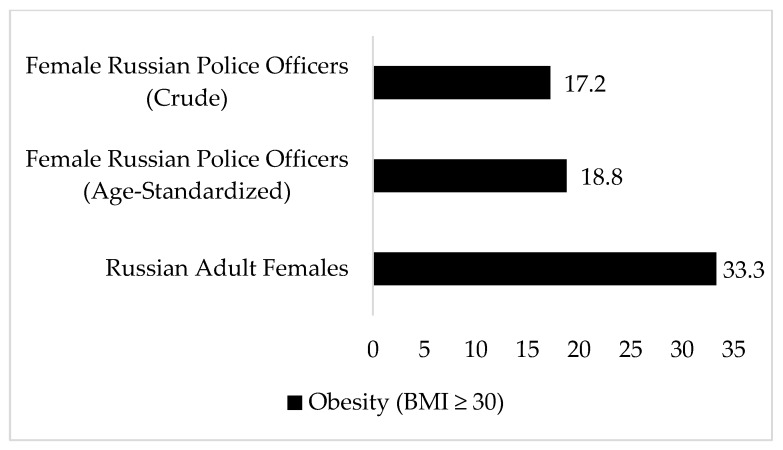
Age-standardized prevalence of obesity (BMI ≥30 kg/m^2^) in female Russian police officers compared with Russian adults. Russian police officer rates were age-standardized using 2014 age-stratified population estimates for Russian females provided in the Central Intelligence Agency World Factbook for Russia [21]; Russian female obesity estimates are from the World Health Organization [22].

**Table 1 ijerph-17-01937-t001:** Rates of false positives and false negatives using BMI-defined obesity classification in comparison to waist circumference (WC)-defined and body fat percentage (BF%)-defined obesity for men (*n* = 153).

Obesity Status	Obesity Status Comparisons		Misclassification Types and Rates
	Waist Circumference		
	Obese (>102 cm)	Non-obese (≤102 cm)	
Obese (BMI ≥ 30.0 kg/m^2^)	2	5	False positives = 3.4%
Non-obese (BMI < 30.0 kg/m^2^)	3	143	False negatives = 0.6%
			Overall diagnostic accuracy = 94.8%
	Body Fat Percentage		
	Obese (≥25%)	Non-obese (<25%)	
Obese (BMI ≥ 30.0 kg/m^2^)	4	3	False positives = 2.5%
Non-obese (BMI < 30.0 kg/m^2^)	30	116	False negatives = 11.8%
			Overall diagnostic accuracy = 78.4%

**Table 2 ijerph-17-01937-t002:** Rates of false positives and false negatives using BMI-defined obesity classification in comparison to WC-defined and BF%-defined obesity for women (*n* = 29).

Obesity Status	Obesity Status Comparisons		Misclassification Types and Rates
	Waist Circumference		
	Obese (>88 cm)	Non-obese (≤88 cm)	
Obese (BMI ≥ 30.0 kg/m^2^)	3	2	False positives = 7.7%
Non-obese (BMI < 30.0 kg/m^2^)	0	24	False negatives = 0%
			Overall diagnostic accuracy = 93.1%
	Body Fat Percentage		
	Obese (≥30%)	Non-obese (<30%)	
Obese (BMI ≥ 30.0 kg/m^2^)	5	0	False positives = 0%
Non-obese (BMI < 30.0 kg/m^2^)	11	13	False negatives = 68.8%
			Overall diagnostic accuracy = 62.1%

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
