# Peer review of "Despite Low Obesity Rates, Body Mass Index Under-Estimated Obesity among Russian Police Officers When Compared to Body Fat Percentage"

_ijerph, 2020, doi:10.3390/ijerph17061937_

Round 1

Reviewer 1 Report

The authors reported that rates of obesity and compare age-standardized sex-specific rates for officers to the adult population in Russia. And they examined the accuracy of BMI-based obesity classification when compared to BF% and waist circumference (WC) measurements.

The most important concerns about this manuscript is that the purpose of the study and the choice of study population are inappropriate.

I could not understand why the authors select the study population as police officer

I think that to examine the  accuracy of BMI-based obesity classification when compared to BF% and waist circumference (WC) measurements have to be applied to general population.

Method

BF% was calculated by dividing fat mass by total body mass and multiplying by 100;

How they measure the fat mass? It should be described in detail.

Disuccison is not sufficient.For example, the explanation about sex-differences are needed.

Results

Line 80-81: These sentences are unnecessary.

Line 94: Rbmi-wc =0.0.46: It is a typo.

Author Response

Please note that our responses are indicated in bold font below.

The authors reported that rates of obesity and compare age-standardized sex-specific rates for officers to the adult population in Russia. And they examined the accuracy of BMI-based obesity classification when compared to BF% and waist circumference (WC) measurements.

The most important concerns about this manuscript is that the purpose of the study and the choice of study population are inappropriate.

I could not understand why the authors select the study population as police officer

We chose to study obesity rates among Russian police officers for 2 key reasons: 1) As we explain in the introduction, obesity among police officers is greater than the general public and even up to 80% in other countries. 2) Our paper directly adds to the literature as there are no published data for obesity measurements among Russian police officers.

I think that to examine the  accuracy of BMI-based obesity classification when compared to BF% and waist circumference (WC) measurements have to be applied to general population.

We agree with the reviewer, but unfortunately general population data only currently include BMI and neither body fat percentage or waist circumference measurements are available. We have added this as a potential limitation in the discussion as follows:

Since only BMI data were available for the Russian population, we were unable to compare obesity rates based on BF% and WC with our sample.

Method

BF% was calculated by dividing fat mass by total body mass and multiplying by 100;

How they measure the fat mass? It should be described in detail.

We reported that we measured fat mass using the Medras tetra-polar bioelectrical impedance now in lines 81-84.

Disuccison is not sufficient.For example, the explanation about sex-differences are needed.

We have added an entire paragraph in the discussion regarding potential explanations for the sex-differences evident in our results section.

Results

Line 80-81: These sentences are unnecessary.

Thank you. We forgot to delete the template instructions. They are now deleted.

Line 94: Rbmi-wc =0.0.46: It is a typo.

Thank you. We have fixed the error.

Reviewer 2 Report

The article presents data on the obesity prevalence among Russian police officers (153 men and 29 women). The diagnosis was based on the WHO classification, as well as criteria for the BF% evaluated by the bio-impedancemetry.

Of note, the division of participants into the presence of obesity according the waist circumference is not correct, because the value >102 cm for men and >88 cm for women, according to the recommendations [10], is used to diagnose abdominal obesity, and not for just obesity. Ref 10, used in the article, recommends these values for the detection of cardiovascular diseases risk in obese or overweight subjects.

In this regard, corrections to the presented data and definitions used by the authors in the materials/methods and results is necessary.

Author Response

The article presents data on the obesity prevalence among Russian police officers (153 men and 29 women). The diagnosis was based on the WHO classification, as well as criteria for the BF% evaluated by the bio-impedancemetry.

Of note, the division of participants into the presence of obesity according the waist circumference is not correct, because the value >102 cm for men and >88 cm for women, according to the recommendations [10], is used to diagnose abdominal obesity, and not for just obesity. Ref 10, used in the article, recommends these values for the detection of cardiovascular diseases risk in obese or overweight subjects.

In this regard, corrections to the presented data and definitions used by the authors in the materials/methods and results is necessary.

We agree with the reviewer that WC is used to diagnose abdominal obesity and is also an indicator of cardiovascular disease risk. However, multiple published research studies have compared measurements between the three measures (BMI, WC, and BF%) including one cited by the CDC (Wohlfahrt-Veje et al., 2014 doi:10.1038/ejcn.2013.28). We have added a paragraph to the introduction to explain the use of WC in comparison to BMI and BF%.

Reviewer 3 Report

This study compared the measurement of obesity among Russian police officers. The authors found that Russian police officers had a lower obesity rate than the general Russian adult population. There was a much higher body fat %-obesity than obesity-based on waist circumference or BMI. They concluded that BMI did not over-estimate obesity in Russian police officers but may underestimate when compared with body fat%. BMI based obesity was more accurate compared to waist circumference however.

Minor comments

  1. Introduction (Line 35) - I think it should be "saturated fats" not excess "unsaturated fats"
  2. Results (Line 80-82) - sounds like instructions - should it be here? Is it a mistake?
  3. Page 3 (Line 94) - small error (rBMI-WC = 0.0.46)
  4. Discussion - was the adult Russian population BMI comparison also age-standardized? If not, and just the Russian police officers were age-standardized BMI the lower BMI in the Russian police officers might be because of the younger age of police officers?

Author Response

Please note that our responses are indicated in bold font below.

This study compared the measurement of obesity among Russian police officers. The authors found that Russian police officers had a lower obesity rate than the general Russian adult population. There was a much higher body fat %-obesity than obesity-based on waist circumference or BMI. They concluded that BMI did not over-estimate obesity in Russian police officers but may underestimate when compared with body fat%. BMI based obesity was more accurate compared to waist circumference however.

Minor comments

  1. Introduction (Line 35) - I think it should be "saturated fats" not excess "unsaturated fats"

We agree and have made that change.

  1. Results (Line 80-82) - sounds like instructions - should it be here? Is it a mistake?

Yes, we have now deleted the template instructions.

  1. Page 3 (Line 94) - small error (rBMI-WC = 0.0.46)

We have fixed the error.

  1. Discussion - was the adult Russian population BMI comparison also age-standardized? If not, and just the Russian police officers were age-standardized BMI the lower BMI in the Russian police officers might be because of the younger age of police officers?

BMI for Russian police officers was standardized to the Russian general population using the direct method. This results in the computation of a value that would reflect the police officers BMI if they had the same age distribution as the Russian population.

Reviewer 4 Report

Introduction:

Overall, the introduction is excessively short it could be beefed up a bit. Why is this topic important? How does fitness levels and BMI and obesity effect police officers? I also think you need to indicate this was an observational study or descriptive study.

Line 42: Why do their job demands require them to have higher levels of fitness levels? This is important because in Line 46 you say they have a sedentary job.

Line 46: You need more than one reference to support this sentence.

Methods:

Line 61: Indicate how many partook in the study.

Line 66. Were there any inclusion and exclusion criteria?

Line 66-67: What were subject’s pre test guidelines? What is the accuracy of the medas tetra-polar BIA? Were subjects properly hydrated? The protocol for the BIA needs to be described.

Line 67: Need to write out Waist Circumference (WC) first.

Line 72: What was the level of significance set too?

Line 76: National rates of what?

Line 77: How were false a positives and negatives

Discussion:

Restate the purpose again

Line 145: So why is this finding important

Author Response

Please note that our responses are indicated in bold font below.

Introduction:

Overall, the introduction is excessively short it could be beefed up a bit. Why is this topic important? How does fitness levels and BMI and obesity effect police officers? I also think you need to indicate this was an observational study or descriptive study.

We have added to the introduction including how obesity is a risk factor for already increased cardiovascular disease risk among police officers. We also added that the study was observational in the study purpose statement.

Line 42: Why do their job demands require them to have higher levels of fitness levels? This is important because in Line 46 you say they have a sedentary job.

We have clarified the relationship between needing to have high fitness levels yet spending most time in sedentary activities in what is now the first paragraph of the introduction.

Line 46: You need more than one reference to support this sentence.

We have clarified that the sentence describes study results from the US state of Iowa.

Methods:

Line 61: Indicate how many partook in the study.

We had provided that information in the results, but we have also added it to the Methods section.

Line 66. Were there any inclusion and exclusion criteria?

Police officers were required to complete an annual medical evaluation that was scheduled in the morning between 0800-0900 hours. Those who slept less than 7 hours the previous night or had eaten prior to their appointment were excluded from the study. We have added this information to the Materials and Methods section.

Line 66-67: What were subject’s pre test guidelines? What is the accuracy of the medas tetra-polar BIA? Were subjects properly hydrated? The protocol for the BIA needs to be described.

Participants were instructed to have an adequate night’s sleep and not consume any food or drinks prior to testing. The Medas 1.0 (SRC Medas, Moscow, Russia), a whole body tetrapolar bioelectrical impedance device that used a 50 kHZ frequency, was used with participants in a supine position using conventional tetrapolar electrode (i.e., Ag/AgCl Schiller bioadhesive electrodes) placement. We have added this information to the Materials and Methods section.

Line 67: Need to write out Waist Circumference (WC) first.

We have written out waist circumference the first time we use it in the introduction.

Line 72: What was the level of significance set too?

We have added that statistical significance was set to p ≤ 0.05 at the end of the paragraph.

Line 76: National rates of what?

We have clarified that this meant national obesity rates based on BMI.

Line 77: How were false a positives and negatives

We are not sure what the reviewer is asking for this comment. We computed both rates (false positives and false negatives) as shown in Tables 1 and 2.

Discussion:

Restate the purpose again

We have added the following in the first discussion paragraph: We compared age-standardized sex-specific obesity rates to the adult Russian population and examined the accuracy of BMI-based obesity classification in comparison to BF% and WC.

Line 145: So why is this finding important

We have added the following sentence to explain: Thus when measuring obesity among Russian police officers, it may be helpful to directly assess BF% rather than using estimation methods such as BMI and WC.

Round 2

Reviewer 1 Report

Thank you for revision.

Reviewer 4 Report

Thank for making the edits.